# Modulation Effects of Repeated Transcranial Direct Current Stimulation at the Dorsolateral Prefrontal Cortex: A Pulsed Continuous Arterial Spin Labeling Study

**DOI:** 10.3390/brainsci13030395

**Published:** 2023-02-25

**Authors:** Valeria Sacca, Nasim Maleki, Ya Wen, Sierra Hodges, Jian Kong

**Affiliations:** Department of Psychiatry, Massachusetts General Hospital and Harvard Medical School, 120 2nd Avenue, Charlestown, MA 02129, USA

**Keywords:** dorsolateral prefrontal cortex (DLPFC), lateral prefrontal cortex (LPFC), pulsed continuous arterial spin labeling (pCASL), transcranial direct current stimulation (tDCS)

## Abstract

Transcranial direct current stimulation (tDCS) is a promising non-invasive method to modulate brain excitability. The aim of this study was to better understand the cerebral blood flow (CBF) changes during and after repeated tDCS at the right dorsolateral prefrontal cortex (DLPFC) in healthy participants using pulsed continuous arterial spin labeling (pCASL). Elucidating CBF changes associated with repeated tDCS may shed light on the understanding of the mechanisms underlying the therapeutic effects of tDCS. tDCS was applied for three consecutive days for 20 min at 2 mA, and MRI scans were performed on day 1 and 3. During anodal tDCS, increased CBF was detected in the bilateral thalamus on day 1 and 3 (12% on day 1 and of 14% on day 3) and in the insula on day 1 (12%). After anodal tDCS on day 1, increased CBF was detected in the cerebellum and occipital lobe (11.8%), while both cathodal and sham tDCS were associated with increased CBF in the insula (11% and 10%, respectively). Moreover, anodal tDCS led to increased CBF in the lateral prefrontal cortex and midcingulate cortex in comparison to the sham. These findings suggest that tDCS can modulate the CBF and different tDCS modes may lead to different effects.

## 1. Introduction

Transcranial direct current stimulation (tDCS) is a promising non-invasive tool for the modulation of cortical excitability [1,2,3]. In recent years, tDCS has been applied as a new treatment option for several neurological and psychiatric disorders with promising results [2,4,5,6,7]. In addition, because tDCS can non-invasively alter the excitability of the targeted brain areas, it could also be used as a research tool to investigate the relationship and interaction between the brain and behavior. However, the underlying mechanisms and widespread effects of this technique are still under investigation, particularly for identifying the impact of tDCS not only on the stimulated area, but also on the brain regions anatomically connected to the targeted brain region.

For this purpose, different brain imaging studies using tDCS in healthy controls and pathological populations have been performed in recent years [2,8,9,10,11,12,13]. These studies suggest that tDCS can modulate brain connectivity and excitability, and the modulation effects vary with different tDCS modes (such as anodal and cathodal) as well as with different targeted brain regions [2,8,9,10,11,12,14]. For example, studies suggest that tDCS applied at the dorsolateral prefrontal cortex (DLPFC) can modulate behavioral and cognitive tasks that involve working memory and can impact conditions such as depression and pain [7,15,16]. In addition, the application of tDCS at the DLPFC can also enhance the placebo effect [17]. Moreover, Zheng et al. [2] found that anodal and cathodal tDCS applied at the right motor region (over C4) can modulate not only the targeted brain area but also a network of brain regions functionally related to the stimulated area. They also found that anodal stimulation induced a larger increase in the cerebral blood flow (CBF) in comparison with cathodal tDCS. In a more recent study, Tu et al. [10] first identified two reoccurring co-activation patterns (CAPs) and calculated their temporal properties (e.g., occurrence rate and transitions) before administering tDCS. Then, they investigated how active tDCS compared to sham tDCS in the modulation of the occurrence rates of these different CAPs and perturbations of transitions between CAPs. The results showed that the occurrence rate of one coactivation pattern (CAP) was significantly decreased by enhancing the excitability of the right DLPFC and left orbitofrontal cortex (lOFC) through the tDCS, while that of another CAP was significantly increased during the first 6 min of stimulation. Furthermore, these tDCS-associated changes persisted over subsequent testing sessions (both during and before/after tDCS) across three consecutive days. Active tDCS could perturb transitions between CAPs and a non-CAP state (when the rDLPFC and lOFC were not activated), but not the transitions within CAPs. These results demonstrated the feasibility of modulating fMRI brain dynamics, which may facilitate the development of new treatments for disorders with altered dynamics.

Arterial spin labeling (ASL) is a magnetic resonance (MR) imaging technique used to assess the cerebral blood flow noninvasively by magnetically labeling inflowing blood [18]. The ASL MRI perfusion produces a “flow labeled image or tag image” and a “control image” in which the static tissue signals are identical, but the magnetization of the inflowing blood is different. In this technique, arterial blood water is magnetically tagged before it enters the tissue of interest. Compared to blood-oxygen-level-dependent (BOLD) imaging, which reflects changes in the oxygenation level, ASL is a direct measure of cerebral blood flow (CBF). Therefore, ASL may be a promising tool for investigating the modulation effects of tDCS [2,8] as it can provide the closest proxy for actual changes in brain activity.

This combined approach of using ASL to measure the effects of tDCS can help researchers and clinicians better understand the underlying neural mechanisms involved in various disorders or pathologies and foster the development of new and more effective therapeutic interventions. Compared to resting state fMRI, one of the main advantages of ASL is that it allows for the measurement of cerebral blood flow changes in response to neural activity. This provides a direct and quantitative measure of the metabolic changes in the brain associated with neural activity, which can be used to identify brain regions that are active during a particular task, stimulation, or cognitive process. This is while functional connectivity-based resting-state fMRI data primarily provide connections between different brain regions.

In a previous study, Stagg et al. 2013 [8] found that anodal tDCS applied at the left DLPFC led to increased perfusion in brain regions closely anatomically connected to the DLPFC in comparison with the cathodal mode. In addition, they found cortical perfusion changes were markedly different during these two time periods, with widespread decreases in cortical perfusion being demonstrated after both anodal and cathodal tDCS compared to the period during stimulation. These results may explain the different effects on behavior in these time periods described previously in the motor system. In a recent study, Shinde et al. [9] applied three tDCS dose levels (Sham, 2 mA, and 4 mA) and two different electrode montages (unihemispheric and bihemispheric) to investigate dose and montage effects on CBF and a finger sequence task. They found changes in the finger sequence task for both hands showing a linear tDCS dose response but no montage effect. The CBF in the right hemispheric perirolandic area increased with dose under the anodal electrode (C4). These results support not only a strong direct tDCS dose effect for CBF and finger sequence task performance as surrogate measures of targeted brain regions but also the indirect effects on rCBF in functionally connected regions, which may allow for the development of new non-invasive treatments for different neurological conditions.

In addition, most of previous studies evaluated the modulation effects considering only one tDCS session using a cross-over design. Thus, there is a lack of knowledge about the possible effects of repeated tDCS sessions that may lead to greater and prolonged modulation effects. Investigating and understanding the CBF changes associated with repeated tDCS sessions may be crucial, because repeated tDCS sessions have been used to induce therapeutic benefits for several psychiatric and neurological disorders, including pain conditions.

In this study, the modulation effects of repeated tDCS on cerebral blood flow were investigated. Healthy participants were recruited and randomized into three groups: anodal, cathodal, and sham tDCS. The hypothesis was that these three different tDCS modalities (stimulation modes) might alter the cerebral blood flow differently and that the modulation effects of the tDCS may involve brain areas structurally and functionally connected with the DLPFC and with important cognitive tasks. Understanding how repeated tDCS may change the CBF in these regions may shed light on the mechanisms underlying the therapeutic effects of tDCS.

This paper is organized in the following sections: (i) Material and Methods for explaining the data collection and analysis; (ii) Results, in which the significant findings are reported; (iii) Discussion, in which the interpretations of these results and their implications are discussed; and (iv) Conclusion.

## 2. Materials and Methods

In this study, healthy participants were recruited and randomized into one of the three tDCS groups. Each participant received tDCS (2 mA) at the right DLPFC over three consecutive days.

### 2.1. Participants

A total of 103 healthy participants without psychiatric and neurologic disorders, based on their reports, were recruited at Massachusetts General Hospital (MGH) from September 2016 to March 2019. The MGH Institutional Review Board (IRB) approved the study with the code 2015P000685, and informed consent was obtained from all the participants.

The original design of the study was to study the modulation effects of tDCS on placebo analgesia and nocebo hyperalgesia [17]. In particular, participants were first trained on how to rate the experimental heat pain stimuli applied on their forearm, then an expectancy manipulation model was used to induce positive and negative expectations with the application of three different inert creams, labeled as: (i) lidocaine for inducing expectations of decreased pain; (ii) capsaicin for the expectations of increased pain; and (iii) neutral moisturizer as a control. After that, the tDCS was applied three times for three consecutive days (day 1–3), and MRI scans were acquired during day 1 and day 3. Placebo and nocebo assessments (through the application of the heat pain stimuli and the three different creams on forearms) were performed at the end of the MRI session (after tDCS) on day 3. More details about the study and demographic parameters of the participants can be found in the previous publication [17]. In addition, the resting state fMRI (BOLD) data acquired before and after the tDCS has been used to analyze the effects of tDCS on brain dynamics [10].

The current study only focused on how anodal, cathodal, and sham tDCS can modulate the cerebral blood flow as measured by ASL. The ASL data in this experiment have never been published before. Since the heat pain stimulation was not applied before or during the collection of the ASL data, the ASL data were not influenced by the application of pain stimulation.

### 2.2. tDCS Modes

The tDCS was applied on three consecutive days at 2 mA for 20 min using the StarStim system (Neuroelectrics, Spain, https://www.neuroelectrics.com/ (accessed on 1 January 2022)). The MRI scans were performed on day 1 and day 3 (Figure 1). Three tDCS modes were used in this study: (1) Anodal tDCS: an anodal electrode at the right dorsolateral prefrontal cortex (rDLPFC, F4 of the 10–20 EEG system), and a cathodal electrode at the left orbitofrontal cortex (lOFC, Fp1 of the 10–20 EEG system); (2) Cathodal tDCS: a cathodal electrode at the rDLPFC, and an anodal electrode at the lOFC; and (3) sham tDCS: the electrode configuration was the same with the two electrodes placed on F4 and Fp1, but the current was applied for just 15 s at the beginning and at the end of the stimulation to simulate stimulation. A double-blinded module was applied to ensure that the tDCS modes were blinded for the operators and participants.

### 2.3. MRI Acquisition

The MRI scans were acquired at MGH Martinos Center for Biomedical Imaging using a 32-channel radiofrequency head coil in a 3 T Siemens scanner. Structural brain images were acquired using a T1-weighted three-dimensional multiecho magnetization-prepared rapid gradient-echo sequence (voxel size: 1 × 1 × 1 mm^3^, repetition time: 2500 ms, echo time: 1.69 ms, slice thickness: 1 mm, flip angle: 7°, and 176 slices). The pulsed continuous arterial spin labeling (pCASL) sequence [19] was used to acquire perfusion weighted images with 2D gradient echo planar imaging (echo time: 15 ms, repetition time: 3800 ms, flip angle: 90°, slice thickness: 5 mm), and a total of 92 volumes were collected.

pCASL scans were performed on the first and third day. In total, 5 different pCASL datasets were acquired: (i) before, during (beginning 7 min after the tDCS started, lasting for 6 min), and after tDCS (about 7 min after the end of tDCS stimulation) on day 1; and (ii) before and during tDCS on day 3. The design of the study is shown in Figure 1.

To apply the tDCS during the ASL scans, MRI-compatible electrodes and the Neuroelectrics Multi-Channel MRI Extension Kit were used to connect the device outside of the MRI room to the participants in the scanner.

### 2.4. MRI Preprocessing

ASL data were analyzed using tools from the FMRIB software library version 6.0.1 (http://www.fmrib.ox.ac.uk/fsl, (accessed on 22 May 2019) [20]). The data processing was performed using FEAT (fMRI Expert Analysis Tool), version 6.0. A standard preprocessing pipeline was applied: (i) motion correction using MCFLIRT [21]; (ii) nonbrain tissue removal using the Brain Extraction Tool (BET; [22]); (iii) spatial smoothing with a Gaussian kernel of 5 mm full width at half-maximum; and (iv) registration to the Montreal Neurological Institute (MNI) standard brain.

### 2.5. Statistical Analysis

To investigate the perfusion or CBF weighted changes induced by tDCS, a mixed-effects analysis was used to examine group-level differences in the perfusion changes for the three contrasts across the whole brain: (i) pre-tDCS vs. during tDCS (days 1 and 3 separately); and (ii) pre-tDCS vs. post-tDCS (day 1). Z-statistic images were thresholded non-parametrically using a cluster-based thresholding to find clusters showing changes in perfusion signals. Clusters were determined by a significance level of Z > 1.64 and a corrected cluster significance threshold of *p* < 0.01.

Comparisons between pre- and during/post-tDCS were performed within the three groups (anodal, cathodal, and sham) independently, and a comparison between the groups (anodal vs. cathodal, anodal vs. sham, and cathodal vs. sham) was performed to assess any possible statistical differences among the three tDCS modalities in the CBF changes between pre- and during/post-tDCS.

## 3. Results

### 3.1. Participants

Eighteen participants dropped at the beginning of the study due to scheduling conflicts or loss of interest, four dropped after randomization due to device error or scheduling issues (one from the sham group, and three from the cathodal). The final cohort used for the analysis was composed of 81 subjects (37 females, mean ± SD age: 27.4 ± 6.4), with 27 participants in each of the three tDCS groups. More details are provided in our previous publication [17]. No statistical differences were found in the age (*p* = 0.84) or gender (χ^2^ = 0.82) between the three groups.

In the second half of enrolled participants, the sensations evoked by the three tDCS modes were evaluated using a questionnaire of sensations related to transcranial electrical stimulation [23]. The sensations evoked by the three tDCS modes were not significantly different (*p* = 0.71 for day 1; *p* = 0.12 for day 2; *p* = 0.23 for day 3).

### 3.2. Within-Group Arterial Spin Labeling Analysis Results

#### 3.2.1. Anodal tDCS

The results of comparing the data from before and during tDCS on day 1 indicated that anodal tDCS led to increased CBF in the bilateral thalamus and the right insula. On day 3, anodal tDCS was associated with increased perfusion in the bilateral thalamus during stimulation compared to pre-stimulation. Further analysis showed that the increase percentage for the two thalamus clusters on day 1 and 3 were 12% on day 1 and 14% on day 3 (Table 1 and Figure 2).

Comparison between pre- and post-tDCS on day 1 showed that anodal tDCS led to increased perfusion in the cerebellum and occipital lobe (increase of 11.8%; Table 1).

#### 3.2.2. Cathodal tDCS

Within-group comparison between pre- and post-tDCS on day 1 showed that cathodal tDCS was associated with increased perfusion in the right insula (11%). No other significant results were found (Table 1 and Figure 2).

#### 3.2.3. Sham tDCS

On day 1, the sham group showed increased perfusion in the right insula after tDCS (10%). No other significant results were found (Table 1 and Figure 2).

#### 3.2.4. Between-Groups Arterial Spin Labeling Analysis Results

The between-group comparisons of the differences of pre- and post-tDCS on day 1 for the three groups showed that anodal tDCS was associated with a greater increase in perfusion in the lateral prefrontal cortex (LPFC) and midcingulate cortex (MCC) compared to sham tDCS on day 1. No other significant results were found (Table 2 and Figure 3). Additionally, no significant differences were detected across the three groups in the changes during tDCS on day 1 and day 3.

## 4. Discussion

In this study, the CBF changes associated with repeated tDCS (anodal, cathodal, and sham) applied at the right DLPFC were investigated to determine whether the mode of tDCS stimulation modulates CBF, as a proxy for brain activity, differently. It was found that: (i) during anodal tDCS on day 1, the CBF significantly increased in the bilateral thalamus and insula, whereas CBF only increased in the bilateral thalamus on day 3. There was also an increase in CBF in the cerebellum and occipital lobe after anodal tDCS treatment on day 1 compared to the pre-tDCS baseline; (ii) the comparison between pre- and post-tDCS on day 1 revealed increased CBF in the right insula in both the cathodal and sham tDCS groups; and (iii) after tDCS treatment (compared to pre-tDCS) on day 1, anodal tDCS was associated with increased CBF in the bilateral MCC and LPFC in comparison to the sham group. These results confirmed that anodal and cathodal tDCS could modulate CBF differently.

The insula and thalamus are anatomically and functionally connected to the DLPFC [24,25,26,27]. Both areas are involved in sensory processing, including the experience of pain [28,29,30,31]. The increased CBF in these two regions indicates that anodal tDCS at the right DLPFC can considerably modulate important regions in the pain pathway.

In a previous study, Stagg et al. [8] found that anodal tDCS at the left DLPFC led to increased CBF in the left primary sensory cortex, midcingulate cortex, paracingulate cortex, and parietal cortex, while CBF decreased in the bilateral thalamus during cathodal tDCS at the left DLPFC. Instead, this study found that CBF increased in the bilateral thalamus during the 6 min scan in the middle of the 20 min anodal tDCS. The different results in the two studies could be due to different reasons: (1) Different target areas: in this study, the right DLPFC was stimulated while Stagg et al. stimulated the left DLPFC. (2) Different intensities of the current applied for the stimulation: this study used 2 mA while Stagg et al. used 1 mA. (3) The results of this study are based on the CBF changes in the middle of the 20 min tDCS treatment (6 min), while the results of Stagg et al. are based on the average changes throughout the 20 min tDCS treatment. These results suggest that locations and intensity may influence the CBF change associated with tDCS.

Additionally, it is worth noting that repeated tDCS stimulation was applied on three different days (MRI scans were performed on day 1 and day 3), and the results showed that CBF increased at the thalamus on both day 1 and 3. Further studies with CBF measurements at multiple time points are needed to further elucidate the patterns of CBF changes across repeated tDCS treatments.

Compared to sham tDCS (day 1), anodal tDCS increased CBF in the bilateral LPFC and MCC (please note, there was no post-tDCS data collected on day 3, thus the reliability of this finding could not be tested). Both the MCC and LPFC are multifunction brain regions. For instance, the literature suggests that the MCC is involved in pain (particularly the affective component of pain), motor function, conflict, error detection, expressing affect and executing goal-directed behavior [32,33], response selection and feedback-guided decision making [34], and processing of information that has the greatest influence on social behavior [35].

Studies also found that the LPFC is involved in cognitive control of motor behavior [36]. In humans with large LPFC damage, the most common symptom is the inability to formulate and carry out plans and sequences of actions including sequences of spoken and written language [37]. The DLPFC is a widely used target of brain stimulation methods for the treatment of multiple disorders, such as chronic pain [38,39,40,41], depression, and other disorders [42]. Taken together, it can be speculated that the increased CBF at the MCC and DLPFC after tDCS could be part of the mechanisms underlying the treatment effects of tDCS for these disorders.

On day 1, after anodal tDCS, increased CBF was also found in the cerebellum and occipital lobe, which are functionally connected to the DLPFC [43,44,45] and are involved in the processing of executive tasks such as language [44] and memory [46,47]. In particular, the occipital lobe is involved in reading/recognizing words and interacts closely with the parietal and temporal lobes to process visual stimuli [48]. To continue, the cerebellum plays a crucial role in balance and motor coordination [49,50,51], and cerebellar damage is mainly correlated with movement dysfunctions [51]. Some studies have shown that the cerebellum is also involved in cognitive/psychiatric disorders such as attention deficit and anxiety disorders, autism spectrum disorder, schizophrenia, and major depressive disorder [49,50]. The results of this study may suggest the potential benefits of using tDCS applied at the right DLPFC for the treatment of these cognitive/psychiatric disorders.

Finally, the results of this study indicate that the concurrent application of tDCS and ASL sequence imaging may be useful for understanding the effects of the stimulation on brain activity and has shown promising results in the study of brain functions. Altering the brain excitability with tDCS and then quantifying the resulting CBF changes with ASL could possibly provide more information on the underlying neural mechanisms involved in various pathological brain disorders, fostering the development of new therapeutic interventions. However, more research studies are needed to fully understand its potential role in the development of new therapies for a wide range of neurological and psychiatric conditions.

It is important to consider some limitations when interpreting the findings of this study. For example, no MRI data were collected during the tDCS session on day 2. As a result, the CBF data on day 2 were unable to be obtained, thus CBF changes on day 2 and whether they were compatible with the findings from day 1 and 3 could not be determined. Furthermore, no pCASL data were collected after tDCS on day 3, thus the post-tDCS effects in day 3 could not be assessed. Finally, further studies should be performed to replicate these results and establish a possible correlation between the CBF changes and clinical outcomes in patient populations.

## 5. Conclusions

This study found that the repeated anodal tDCS applied on the right DLPFC may modulate the CBF in brain regions involved in pain perception and modulation (such as the thalamus, insula, LPFC, and MCC) as well as the regulation of visual information processing (such as the occipital lobe) and balance and motor coordination (such as the cerebellum). These findings suggest that different tDCS modalities may be associated with different changes in CBF during and after tDCS. Understanding the CBF alterations associated with different tDCS modes may provide insights into the potential therapeutic effects of tDCS.

## Figures and Tables

**Figure 1 brainsci-13-00395-f001:**
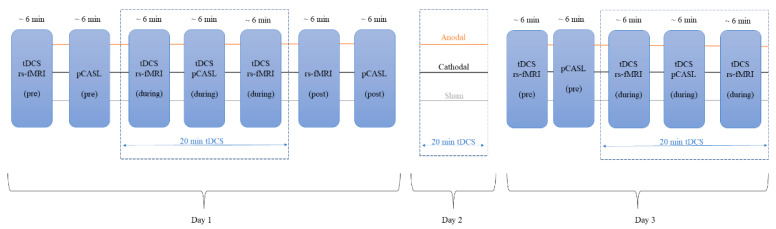
Experimental design: 20 min of anodal, cathodal, and sham tDCS was applied on 3 consecutive days. On day 1, pCASL sequences were collected before the tDCS (pCASL (pre)), during the tDCS (beginning ~6 min into the stimulation; tDCS pCASL (during)), and after the tDCS (pCASL (post)). On day 2, the participants received only the tDCS treatment, and no MRI data were acquired. On day 3, pCASL scans were acquired before the tDCS (pCASL (pre)), and during the tDCS (beginning ~6 min into the stimulation; tDCS pCASL (during)). rs-fMRI: resting-state fMRI; pCASL: pulsed continuous arterial spin labeling.

**Figure 2 brainsci-13-00395-f002:**
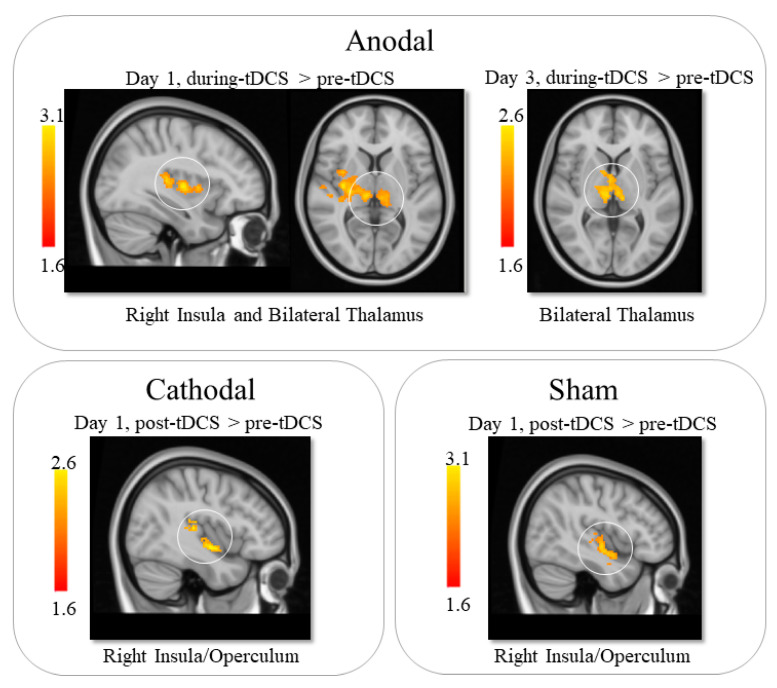
ASL statistical analysis results. Anodal tDCS led to increased CBF in the bilateral thalamus and insula during stimulation on day 1. While on day 3, anodal tDCS was associated with increased CBF in the bilateral thalamus during stimulation. Cathodal and sham tDCS led to increased CBF in the right insula after stimulation on day 1.

**Figure 3 brainsci-13-00395-f003:**
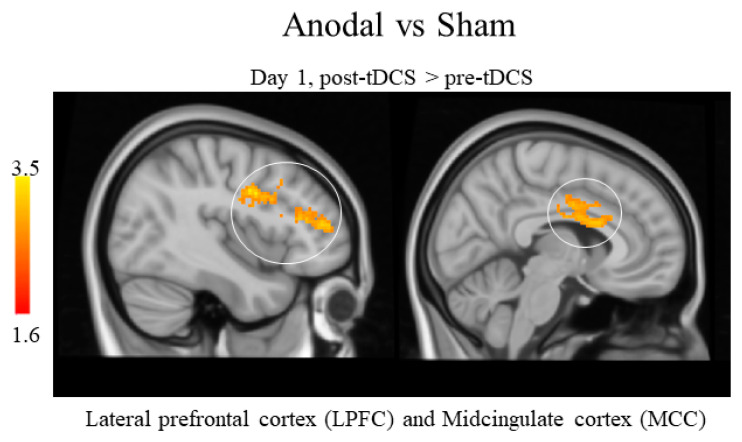
ASL statistical analysis between groups comparison. Anodal tDCS led to increased CBF in the lateral prefrontal cortex (LPFC) and midcingulate cortex (MCC) compared to sham tDCS after stimulation on day 1.

**Table 1 brainsci-13-00395-t001:** ASL statistical analysis within the three group comparisons. The threshold was set at *p* < 0.05 and Z > 1.64.

Comparison	Region	Peak MNI Coordinates		
		x	y	z
**Anodal tDCS**				
Day 1—pre vs. during tDCS	Left Thalamus	−10	−26	10
Day 1—pre vs. during tDCS	Right Thalamus	14	−26	14
Day 1—pre vs. during tDCS	Right Insula	50	−24	8
Day 1—pre vs. post tDCS	Cerebellum	−8	−58	−14
Day 1—pre vs. post tDCS	Occipital Lobe	18	−90	20
Day 3—pre vs. during tDCS	Bilateral Thalamus	−10	−22	16
**Cathodal tDCS**				
Day 1—pre vs. post tDCS	Right Insula	38	−22	0
**Sham tDCS**				
Day 1—pre vs. post tDCS	Right Insula	50	−4	−8

**Table 2 brainsci-13-00395-t002:** ASL statistical analysis between the three group comparisons. The threshold was set at *p* < 0.05 and Z > 1.64.

Comparison	Region	Peak MNI Coordinates		
		x	y	z
**Anodal vs. Sham**				
Day 1—pre vs. post tDCS	Bilateral Middle Cingulate Cortex	−12	4	30
Day 1—pre vs. post tDCS	Bilateral Lateral Prefrontal Cortex	38	34	20
**Cathodal vs. Sham**				
No significant results were found				
**Anodal vs. Cathodal**				
No significant results were found				

## Data Availability

The data that support the findings of this study are available from the corresponding author upon reasonable request.

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
