# Peer review of "Modulation Effects of Repeated Transcranial Direct Current Stimulation at the Dorsolateral Prefrontal Cortex: A Pulsed Continuous Arterial Spin Labeling Study"

_brainsci, 2023, doi:10.3390/brainsci13030395_

Round 1

Reviewer 1 Report

The paper is interesting, relevant, and well-written. The line number insertion along the manuscript would be helpful to provide the review. Just minor reviews are necessary. As the thalamus and insula were the ASL trend marks with tDCS in DLPFC. Even the sensory processing be related in the discussion, as a suggestion, to this and the next studies, the limbic activation (emotional pain) can be triggered. In this sense, electrodermal activity and cardiac frequency can be used to register (indirectly) the limbic activation, because the simple electrode placement can active the limbic system (that can explain your phrase “On day 1, the sham group showed increased perfusion in the right insula after tDCS”). 

Major

-None.

Minor

-Introduction: in the last paragraph, the phrase: “Each participant received tDCS (2 mA) at the right DLPFC over three consecutive days. Pulsed continuous arterial spin labeling (pCASL) data were collected before (pre-) and during tDCS on the first and third day, and post-tDCS pCASL data was acquired after the treatment on the first day.” may be removed, because this is a method and not an introduction. The hypothesis can be saved. 

-Material and methods: In section “2.3. MRI acquisition”, in the fourth line, in the “mm3”, the “3” must be in superscript. Moreover, sometimes as in “3T Siemens scanner” the unit is attached to the number, other times as in “1 mm” the unit is spaced from the number. This must be standardized throughout the manuscript.    

Author Response

Reviewer 1:

The paper is interesting, relevant, and well-written. The line number insertion along the manuscript would be helpful to provide the review. Just minor reviews are necessary. As the thalamus and insula were the ASL trend marks with tDCS in DLPFC. Even the sensory processing be related in the discussion, as a suggestion, to this and the next studies, the limbic activation (emotional pain) can be triggered. In this sense, electrodermal activity and cardiac frequency can be used to register (indirectly) the limbic activation, because the simple electrode placement can active the limbic system (that can explain your phrase “On day 1, the sham group showed increased perfusion in the right insula after tDCS”). 

  • Authors’ response: We thank the reviewer for their valuable comments and suggestions for future studies with the introduction of the limbic system.

Major

-None.

Minor

-Introduction: in the last paragraph, the phrase: “Each participant received tDCS (2 mA) at the right DLPFC over three consecutive days. Pulsed continuous arterial spin labeling (pCASL) data were collected before (pre-) and during tDCS on the first and third day, and post-tDCS pCASL data was acquired after the treatment on the first day.” may be removed, because this is a method and not an introduction. The hypothesis can be saved. 

  • Authors’ response: We thank the reviewer for this comment, and we have modified this part of the manuscript.

-Material and methods: In section “2.3. MRI acquisition”, in the fourth line, in the “mm3”, the “3” must be in superscript. Moreover, sometimes as in “3T Siemens scanner” the unit is attached to the number, other times as in “1 mm” the unit is spaced from the number. This must be standardized throughout the manuscript. 

  • Authors’ response: We thank the reviewer for this suggestion, and we have standardized the unit on the manuscript.

Reviewer 2 Report

The authors present the paper entitled “Modulation effects of repeated transcranial direct current stimulation at the dorsolateral prefrontal cortex: a pulsed continuous arterial spin labeling study”

This paper investigates the CBF changes associated with repeated tDCS (an-odal, cathodal, and sham) applied at the right DLPFC to determine whether the mode of tDCS stimulation modulates CBF as a proxy for brain activity differently.

The article presents the following concerns:

  • Improve the Abstract section by highlight the purpose of the study. Also, quantitative values must be added in this section.

  • I suggest to improve the objective of the work in the last paragraph of Section 1. Please mention the main controversies and higlight the contributions of the work.

  • Figure 1 must be vectorized in order to see details.

  • “which has never been published before” This sentence is ambiguous. It is recommended to search more recent literature. For example, at the beginning of the manuscript, it is mentioned “In recent years, tDCS has been applied as a new treatment option for several neurological and psychiatric disorders with promising results” but the supported references are from 1964 to 2012. What is the novelty of the work?

  • Table 1: Please describe how the comparison will be made in Section 2. 

  • “day 3) and found that CBF increased at the thalamus on both day 1 & 3” how much increased? 

  • Add a nomenclature table at the end of the manuscript.

  • Add hyperlinks to tables, figures, and references.

  • It is recommendable to describe the structure of the text at the end of the introduction

  • It is recommendable to make a little introduction between points 2 and 2.1, 3 and 3.1

  • The text must be written in the 3rd person or passive voice.

  • The literature should be updated by using up-to-date references. More than 50% is from ten years ago.

  • I recommend decreasing the 53% of similitude trow by Turnitin. 15-20% could be better.

The following misspelling should be checked:

  1. page 2: The use of “and/or” is severely frowned upon in formal writing. Consider using only one conjunction or rewriting the sentence.

  2. page 6: “treatment (6 minutes), while Stagg et al…” The abreviation “et al” seems to be incorrectly punctuated. Consider changing by “et al.”

Author Response

Reviewer 2:

The authors present the paper entitled “Modulation effects of repeated transcranial direct current stimulation at the dorsolateral prefrontal cortex: a pulsed continuous arterial spin labeling study”.

This paper investigates the CBF changes associated with repeated tDCS (anodal, cathodal, and sham) applied at the right DLPFC to determine whether the mode of tDCS stimulation modulates CBF as a proxy for brain activity differently.

The article presents the following concerns:

  • Improve the Abstract section by highlight the purpose of the study. Also, quantitative values must be added in this section.
  • Authors’ response: We thank the reviewer for this valuable comment. We have revised the abstract based on the comments of the reviewers.

  • I suggest to improve the objective of the work in the last paragraph of Section 1. Please mention the main controversies and higlight the contributions of the work.
  • Authors’ response: We have revised the manuscript as suggested (Section 1).

  • Figure 1 must be vectorized in order to see details.
  • Authors’ response: We have revised the manuscript and provided a high resolution figure.

  • “which has never been published before” This sentence is ambiguous. It is recommended to search more recent literature. For example, at the beginning of the manuscript, it is mentioned “In recent years, tDCS has been applied as a new treatment option for several neurological and psychiatric disorders with promising results” but the supported references are from 1964 to 2012. What is the novelty of the work?
  • Authors’ response: We have revised the sentences to avoid ambiguity. We also updated the references on this topic and clarified the novelty of the manuscript.

  • Table 1: Please describe how the comparison will be made in Section 2. 
  • Authors’ response: We have added the description of the comparisons in the statistical analysis section.

  • “day 3) and found that CBF increased at the thalamus on both day 1 & 3” how much increased? 
  • Authors’ response: To answer the question, we extracted the increase percentage for the thalamus on day 1 and 3. We found an increase of 12% on day 1 (effect size: 0.65) and 14% on day 3 (effect size: 0.57).

  • Add a nomenclature table at the end of the manuscript.
  • Authors’ response: We have added the nomenclature table at the end of the manuscript.

  • Add hyperlinks to tables, figures, and references.
  • Authors’ response: We have added the hyperlinks to tables and figures. We will make additional adjustments based on the rules of the journal.

  • It is recommendable to describe the structure of the text at the end of the introduction
  • Authors’ response: We have provided the structure of the text at the end of the introduction.

  • It is recommendable to make a little introduction between points 2 and 2.1, 3 and 3.1
  • Authors’ response: We have added a small introduction paragraph between points 2 and 2.1.

  • The text must be written in the 3rd person or passive voice.
  • Authors’ response: We have revised the manuscript based on this suggestion.

  • The literature should be updated by using up-to-date references. More than 50% is from ten years ago.
  • Authors’ response: We have updated the references.

  • I recommend decreasing the 53% of similitude trow by Turnitin. 15-20% could be better.
  • Authors’ response: We have tried to reduce the similarity. However, there are a lot of technical terms, such as fMRI and tDCS data collection procedures that are standard across studies. We may not be able to reduce it to 15-20% as suggested.

The following misspelling should be checked:

  1. page 2: The use of “and/or” is severely frowned upon in formal writing. Consider using only one conjunction or rewriting the sentence.
  • Authors’ response: We have revised this part of the manuscript.

  1. page 6: “treatment (6 minutes), while Stagg et al…” The abreviation “et al” seems to be incorrectly punctuated. Consider changing by “et al.”
  • Authors’ response: We have revised this part of the manuscript.

Reviewer 3 Report

-Page 2. “Therefore, ASL may be a promising tool for investigating the modulation effects of tDCS (Stagg et al., 2013; Zheng et al., 2011)”. The referencing should be written according to journal guidelines used in the rest of the text. 

- Page 2 “We hypothesized that different tDCS modalities might alter the CBF differently.” Please rewrite to be more understandable for the readers what is meant by “modalities”. 

-Page 3 “mm3”? 

-The tables should be presented according to journal guidelines, removing bold.

-In figure 3, please remove the extra full stop. 

- “Page 6- 1) Different target areas: we stimulated the right DLPFC while they stimulated the leftDLPFC.” Please rewrite to be more clear, instead of “they” it is suggested to mention the first author of the study the authors are referring to. 

-“The dorsal lateral prefrontal cortex is a widely used target of brain stimulation methods for the treatment of multiple disorders, such as chronic pain [33–36], depression, and other disorders [37]”. Please carefully check the usage of acronyms in the manuscript text ( for example, DLPFC) and use them consistently. 

-Regarding reference 39 and language brain areas, do the authors mean to the left or right hemispheric areas devoted to language, and which language components are they referring? 

- Page 7, “Our results may suggest the potential of using tDCS at the DLPFC for treatment of these disorders.” The authors are suggested to reformulate this sentence to be more understandable. Do the authors refer to the results of tDCS for the cerebellum in this sentence? 

- Please highlight novelty and future work more clearly.

Author Response

Reviewer 3:

-Page 2. “Therefore, ASL may be a promising tool for investigating the modulation effects of tDCS (Stagg et al., 2013; Zheng et al., 2011)”. The referencing should be written according to journal guidelines used in the rest of the text.

  • Authors’ response: We thank the reviewer for the comment. We updated these references in the text.

- Page 2 “We hypothesized that different tDCS modalities might alter the CBF differently.” Please rewrite to be more understandable for the readers what is meant by “modalities”.

  • Authors’ response: We changed this sentence in the manuscript.

-Page 3 “mm3”?

  • Authors’ response: We changed mm3 in the text.

-The tables should be presented according to journal guidelines, removing bold.

  • Authors’ response: We have modified the tables.

-In figure 3, please remove the extra full stop.

  • Authors’ response:  We have removed the extra stop.

- “Page 6- 1) Different target areas: we stimulated the right DLPFC while they stimulated the leftDLPFC.” Please rewrite to be more clear, instead of “they” it is suggested to mention the first author of the study the authors are referring to.

  • Authors’ response: We have changed this part of the manuscript.

-“The dorsal lateral prefrontal cortex is a widely used target of brain stimulation methods for the treatment of multiple disorders, such as chronic pain [33–36], depression, and other disorders [37]”. Please carefully check the usage of acronyms in the manuscript text ( for example, DLPFC) and use them consistently.

  • Authors’ response: We have modified this part of the manuscript.

-Regarding reference 39 and language brain areas, do the authors mean to the left or right hemispheric areas devoted to language, and which language components are they referring?

  • Authors’ response: We thank the reviewer for this comment. Literature showed language-related functions of the DLPFC include several different aspects such as discourse management, integration of prosody, interpretation of nonliteral meanings, inference making, ambiguity resolution, and error repair. The DLPFC may be a key region for improving the functional connectivity between the language network and other functional networks, including the subcortical circuits [1]. It seems that both hemispheric areas of the DLPFC play a role in this case. In particular the right side may play an important role for language control and switching in bilingual speakers [2].

[1] Hertrich, I.; Dietrich, S.; Blum, C.; Ackermann, H. The Role of the Dorsolateral Prefrontal Cortex for Speech and Language Processing. Front Hum Neurosci 2021, 15.

[2] Dongxu Liu, Guangyan Dai, Churong Liu, Zhiqiang Guo, Zhiqin Xu, Jeffery A Jones, Peng Liu, Hanjun Liu, Top–Down Inhibitory Mechanisms Underlying Auditory–Motor Integration for Voice Control: Evidence by TMS, Cerebral Cortex, Volume 30, Issue 8, August 2020, Pages 4515–4527, https://doi.org/10.1093/cercor/bhaa054

- Page 7, “Our results may suggest the potential of using tDCS at the DLPFC for treatment of these disorders.” The authors are suggested to reformulate this sentence to be more understandable. Do the authors refer to the results of tDCS for the cerebellum in this sentence?

  • Authors’ response: We thank the reviewer for this valuable suggestion, and we have modified this sentence in the manuscript.

- Please highlight novelty and future work more clearly.

  • Authors’ response: We have included information about the novelty of the work in the Introduction (Section 1) and about the future work in the Discussion (Section 4).

Reviewer 4 Report

1.     The title should mention the article type.

2.     The abstract needs more quantitative data. Please, provide more crude (percentages and numbers) data.

3.     The authors should avoid grouped references. E.g., [2,4–7]; [19–22]; [23–26]; [33–36]; [44–46].

4.     Could the authors be more specific in what their study brings new to the literature? To the reviewer’s knowledge, these queries were already answered by other studies.

5.     Please, provide the IRB number. Including it in the methodology and the specific section designed for the IRB number is recommended. It is worth remembering that specific studies should have specific numbers. The exception is if the project is mentioned the production of more than one manuscript.

6.     Statistics

a.     How was calculated the power of the study?

b.     How were variables distributed?

c.     How were confounding variables assessed?

d.     What was the statistical software used for analysis?

Author Response

Reviewer 4:

  1. The title should mention the article type.
  • Authors’ response: We have inserted the article type in the title.
  1. The abstract needs more quantitative data. Please, provide more crude (percentages and numbers) data.
  • Authors’ response: We have inserted the quantitative data in the abstract.
  1. The authors should avoid grouped references. E.g., [2,4–7]; [19–22]; [23–26]; [33–36]; [44–46].
  • Authors’ response: We thank the reviewer for the suggestion. We are using the reference style suggested from the journal in which grouped references are allowed.
  1. Could the authors be more specific in what their study brings new to the literature? To the reviewer’s knowledge, these queries were already answered by other studies.
  • Authors’ response: We thank the reviewer for this comment. We added this part on the introduction: “Nevertheless, few studies have investigated the CBF changes associated with tDCS, and most of them evaluated the modulation effects considering only one tDCS session using a cross-over design. Thus, there is a lack of knowledge about the effects of repeated tDCS sessions that may lead to a extended deeper modulation effects of the brain activity. In particular, understanding CBF changes associated with repeated tDCS may be crucial and meaningful in the neuroscience field since repeated tDCS may induce prolonged changes in the brain activity and connectivity and has been used the potential to induce therapeutic benefits for several psychiatric and neurologi-cal disorders, including pain conditions. In this study, the modulation effects of re-peated tDCS on CBF were investigated. Healthy participants were recruited and ran-domized into three groups: anodal, cathodal, and sham tDCS. The hypothesis was that these three different tDCS modalities (stimulation modes) might alter the CBF differ-ently.”
  1. Please, provide the IRB number. Including it in the methodology and the specific section designed for the IRB number is recommended. It is worth remembering that specific studies should have specific numbers. The exception is if the project is mentioned the production of more than one manuscript.
  • Authors’ response: We added the IRB number in the Methods section.
  1. Statistics
  2. How was calculated the power of the study?

      Authors’ response: Brain imaging studies do not routinely provide power analysis. For instance, in a recent study on the sample size of brain imaging studies, the authors found that only 4 of 131 papers in top neuroimaging journals in 2017 and 5 of 142 papers in 2018 had pre-study power calculations, most for single t-tests and correlations [1]. In another study, Thirion et al. examined the question of sample sizes for fMRI and found that 20-25 subjects were necessary to achieve reliable results [2].

In a previous study on estimating means and variances in the gray matter, CBF values (36.4 ± 6.40 (mean ± standard deviation) ml/100gr/min) were estimated using pCASL in a single site sample of 17 healthy subjects (9 male) with a comparable age range to ours (between 20 to 29 years) [3] and estimates of perfusion variability of < 20% over repeated measurements 1 to 3 weeks apart within the same six subjects on a 3T scanner [4]. Assuming > 15% difference between the baseline scan and a follow up scan on the same subjects we had > 85% power (alpha = 0.05, two sided) to detect differences in pre and post intervention measurements in a sample size of 27 subjects per group. In our present study, 27 participants in each of the three tDCS groups were included in final analysis. The sample size is similar or greater than previous tDCS ASL studies [5,6]. Thus, we believe this is a sufficiently powered study.

[1] Szucs, D., & Ioannidis, J. P. (2020). Sample size evolution in neuroimaging research: An evaluation of highly-cited studies (1990–2012) and of latest practices (2017–2018) in high-impact journals. NeuroImage, 221, 117164.

[2] Thirion B, Pinel P, Roche A, Dehaene S, Poline J. Analysis of fMRI Data Sampled From Large Populations: Statistical and Methodological Issues. NeuroImage 2007; 35: 105–20.

[3] Alisch, J. S., Khattar, N., Kim, R. W., Cortina, L. E., Rejimon, A. C., Qian, W., ... & Bouhrara, M. (2021). Sex and age-related differences in cerebral blood flow investigated using pseudo-continuous arterial spin labeling magnetic resonance imaging. Aging (Albany NY), 13(4), 4911.

[4] Gevers, S., Van Osch, M. J., Bokkers, R. P., Kies, D. A., Teeuwisse, W. M., Majoie, C. B., ... & Nederveen, A. J. (2011). Intra-and multicenter reproducibility of pulsed, continuous and pseudo-continuous arterial spin labeling methods for measuring cerebral perfusion. Journal of Cerebral Blood Flow & Metabolism, 31(8), 1706-1715.

[5] Stagg, C.J.; Lin, R.L.; Mezue, M.; Segerdahl, A.; Kong, Y.; Xie, J.; Tracey, I. Widespread Modulation of Cerebral Perfusion Induced during and after Transcranial Direct Current Stimulation Applied to the Left Dorsolateral Prefrontal Cortex. Journal of Neuroscience 2013, 33, 11425–11431, doi:10.1523/JNEUROSCI.3887-12.2013.

[6] Zheng, X.; Alsop, D.C.; Schlaug, G. Effects of Transcranial Direct Current Stimulation (TDCS) on Human Regional Cerebral Blood Flow. Neuroimage 2011, 58, 26–33, doi:10.1016/j.neuroimage.2011.06.018.

  1. How were variables distributed?
  • Authors’ response: Linear regression results are generally not sensitive to departure from the assumption of normality, except possibly for very small sample sizes, which is not the case of our study with 27 per group. This is because of the central limit theorem, which ensures that, in most cases, the sampling distributions of averages rapidly approach that of a normal distribution, as the sample size increases. Linear regression estimates are essentially weighted averages of the data, so this result applies not just to t-tests, but to regression in general. This issue is discussed extensively in [7].  Box, Hunter, and Hunter in [8] show graphically that the central limit approach to normality can be substantial even for samples of five or less. 

         [7] Miller, R.G., Jr. (1997). Beyond ANOVA: Basics of Applied             Statistics. Chapman and Hall, London.

         [8] Box, G.E.P, Hunter W.G., Hunter J.S. (1978). Statistics for             Experimenters: An Introduction to Design, Data Analysis,                 and Model Building. John Wiley and Sons, New York.

  1. How were confounding variables assessed?
  • Authors’ response: In this study, no statistical differences were found in the age (p=0.84) or gender (χ2=0.82) between the three groups based on ANOVA and chi-squared test. Thus, the demographic parameters are well balanced among the three groups.
  1. What was the statistical software used for analysis?
  • Authors’ response: The software used for the statistical analysis was FSL with the toolboxes FEAT and GLM.

Round 2

Reviewer 2 Report

The manuscript reached my expectations 

Author Response

1) The manuscript reached my expectations

  • Author’s response: We thank the reviewer for the comment.

Reviewer 4 Report

1.     The Reviewer was not able to observe the article type in the title. E.g., literature review? Case report? Observational study? Cross-sectional study?

2.     “The abstract needs more quantitative data. Please, provide more crude (percentages and numbers) data.” The reviewer would recommend including more information about the CBF in the described regions.

Author Response

1) The Reviewer was not able to observe the article type in the title. E.g., literature review? Case report? Observational study? Cross-sectional study?

  • Author’s response: The article type is Brain Imaging study.

2) “The abstract needs more quantitative data. Please, provide more crude (percentages and numbers) data.” The reviewer would recommend including more information about the CBF in the described regions.

  • Author’s response: We have added the percentages of the CBF changes also for the other regions.
